# SARS-CoV-2 Infection of Lung Epithelia Leads to an Increase in the Cleavage and Translocation of RNase-III Drosha; Loss of Drosha Is Associated with a Decrease in Viral Replication

**DOI:** 10.3390/genes16101239

**Published:** 2025-10-20

**Authors:** Michael T. Winters, Emily S. Westemeier-Rice, Travis W. Rawson, Kiran J. Patel, Gabriel M. Sankey, Maya Dixon-Gross, Olivia R. McHugh, Nasrin Hashemipour, McKenna L. Carroll, Isabella R. Wilkerson, Ivan Martinez

**Affiliations:** 1Department of Microbiology, Immunology and Cell Biology, West Virginia University School of Medicine, West Virginia University, Morgantown, WV 26506, USA; mtwinters@mix.wvu.edu (M.T.W.); twr0001@mix.wvu.edu (T.W.R.); gms00019@mix.wvu.edu (G.M.S.); orm0004@mix.wvu.edu (O.R.M.); irw00003@mix.wvu.edu (I.R.W.); 2West Virginia University Cancer Institute, West Virginia University, Morgantown, WV 26506, USA; emily.rice@emory.edu (E.S.W.-R.); nh00053@mix.wvu.edu (N.H.); mlc00064@mix.wvu.edu (M.L.C.); 3West Virginia University School of Medicine, West Virginia University, Morgantown, WV 26506, USA; kjp0018@mix.wvu.edu; 4Department of Biology, West Virginia University, Morgantown, WV 26506, USA; mrd00026@mix.wvu.edu

**Keywords:** SARS-CoV-2, COVID-19, RNase-III Drosha, miRNA biogenesis, coronavirus, host defense, pathogenic mechanism

## Abstract

**Background/Objectives:** Since its emergence, COVID-19—caused by the novel coronavirus SARS-CoV-2—has affected millions globally and led to over 1.2 million deaths in the United States alone. This global impact, coupled with the emergence of five new human coronaviruses over the past two decades, underscores the urgency of understanding its pathogenic mechanisms at the molecular level—not only for managing the current pandemic but also preparing for future outbreaks. Small non-coding RNAs (sncRNAs) critically regulate host and viral gene expression, including antiviral responses. Among the molecular regulators implicated in antiviral defense, the microRNA-processing enzyme Drosha has emerged as a particularly intriguing factor. In addition to its canonical role, Drosha also exerts a non-canonical, interferon-independent antiviral function against several RNA viruses. **Methods:** To investigate this, we employed q/RT-PCR, Western blot, and immunocytochemistry/immunofluorescence in an immortalized normal human lung/bronchial epithelial cell line (NuLi-1), as well as a human colorectal carcinoma Drosha CRISPR knockout cell line. **Results:** In this study, we observed a striking shift in Drosha isoform expression following infection with multiple SARS-CoV-2 variants. This shift was absent following treatment with the viral mimetic poly (I:C) or infection with other RNA viruses, including the non-severe coronaviruses HCoV-OC43 and HCoV-229E. We also identified a distinct alteration in Drosha’s cellular localization post SARS-CoV-2 infection. Moreover, Drosha ablation led to reduced expression of SARS-CoV-2 genomic and sub-genomic targets. **Conclusions:** Together, these observations not only elucidate a novel aspect of Drosha’s antiviral role but also advance our understanding of SARS-CoV-2 host–pathogen interactions, highlighting potential therapeutic avenues for future human coronavirus infections.

## 1. Introduction

Since its emergence in 2019, the novel coronavirus SARS-CoV-2, the causative agent of COVID-19, has caused more than 1.23 million deaths (as of 24 August 2025) [1]. While members of the Coronaviridae family share a common mechanism of cellular entry, they differ widely in pathogenicity, host range, and genomic organization. Notably, many human coronaviruses (HCoVs) have only been identified in the past 15 years [2]. Four of the seven known human coronaviruses, including HCoV-OC43, HCoV-229E, HKU1, and HCoV- NL63 are known to cause mild disease in humans. Three are known to cause severe disease, and these include MERS-CoV, SARS-CoV-1 and SARS-CoV-2. Coronaviruses, including SARS-CoV-2, possess a single-stranded, positive-sense RNA genome of approximately 30 kb. However, the extent to which it interacts with host non-coding RNA processing machinery remains poorly understood [3].

Host innate and intracellular responses to viral infection are regulated by various mechanisms, including microRNA (miRNA)-mediated translational regulation [4,5]. MiRNAs are short (~22 nt), non-coding RNAs that target mRNAs, promoting their degradation or repressing their translation [6,7]. Several studies have revealed direct interactions between host miRNAs and viral RNA genomes, which facilitate viral replication and aid in the evasion of host immune defenses [4,8]. Moreover, specific miRNAs are critical regulators of innate immune responses, particularly the interferon pathway, and are often dysregulated during viral infections [9,10,11].

Recent studies suggest that RNA-binding proteins and non-coding RNA processing pathways contribute to antiviral defense [2,12]. Additionally, several viruses, including SARS-CoV-2, have been shown to encode viral miRNAs, which interact with host miRNAs to modulate viral replication [13,14,15,16,17]. The biogenesis of miRNAs involves a defined set of processing proteins. This process begins in the nucleus with DGCR8 binding to the hairpin structure of primary miRNAs (pri-miRNAs), followed by cleavage mediated by the RNase III enzyme Drosha [18,19].

Beyond its canonical role, Drosha also functions through non-canonical pathways, including direct binding to viral RNA genomes [20]. During viral infection, Drosha has been shown to translocate from the nucleus to the cytoplasm in response to cellular stress independently of interferon signaling, a pathway known to be suppressed following SARS-CoV-2 infection [21,22,23,24]. Interestingly, cytoplasmic Drosha isoforms may arise via cleavage or alternative splicing [25,26,27].

Previous work from our laboratory has demonstrated both cytoplasmic translocation and isoform variations in Drosha [28]. In this study, we found that SARS-CoV-2 infection induces Drosha translocation to the cytoplasm and preferentially increases the expression of the P140 Drosha isoform—an effect not observed during infections with non-severe coronaviruses. Furthermore, Drosha ablation significantly decreased the expression of several SARS-CoV-2 genomic and sub-genomic targets. Collectively, these findings reveal a unique expression pattern of Drosha isoforms in SARS-CoV-2 infection and suggest the potential role of these isoforms in viral replication.

## 2. Materials and Methods

### 2.1. RNA Extraction and Quantitative Reverse Transcription PCR

TRIzol/Phenol–Chloroform extraction was utilized to isolate total RNA from cell pellets using TRIzol Reagent (Ambion, Austin, TX, USA; 15596026) and chloroform (Fisher Scientific, Waltham, MA, USA; AAJ67241AP), following the standard manufacturer protocol. This was preceded by isopropanol (Fisher Scientific; BP2618500) precipitation with GlycoBlue (Invitrogen, Waltham, MA, USA; AM9516) co-precipitant, RNA pellet wash using 75% ethanol (Fisher Scientific; BP28184), and RNA resuspension in nuclease-free water (Ambion, Austin, TX, USA; AM9932). Removal of potential genomic DNA contaminant was performed using Turbo DNAfree DNase (Invitrogen; AM1907) for 8 min at 37 °C. RNA concentration and purity were assessed using a NanoDrop 2000 Spectrophotometer, with acceptable 260/280 absorbance ratios ranging between 1.8 and 2.1. 1 µg of total RNA was converted to cDNA using the iScript cDNA Synthesis Kit (Bio-Rad, Hercules, CA, USA; 170-6891) and followed by Quantitative PCR using the Sso-Advanced Universal SYBR Green Supermix (Bio-Rad; 172-5271) and forward/reverse primer sets specific to the target genes of interest for this study Appendix A.

Melt profiles were generated for each primer set to ensure a single PCR amplicon, within 2 °C of the predicted melt temperature (TM), was produced. Relative target expression was determined using the ∆∆CT Method: [Relative Expression = 2-∆CT; where CT = Cycle Threshold; where ∆CT = CT (Target RNA)–CT (Endogenous Control RNA)], where the Endogenous Control for mRNA from the cell lines used in this study was GAPDH (glyceraldehyde-3-phosphate dehydrogenase) and/or UBC (Ubiquitin-C).

PCR products were visualized via gel electrophoresis, using 1% agarose (Sigma-Aldrich, St. Louis, MO, USA; A9539) gels (*w*/*v* in 1× Tris/Borate/EDTA buffer) containing 0.5 µg/mL Ethidium Bromide (Sigma-Aldrich; E7637), at 10 V/cm^2^ for 2h. Gels were imaged using an iBright image system/analysis software (Version 1.8.1).

### 2.2. Cell Culture

Cell line characteristics and information can be found in Appendix A. NuLi-1 cells (ATCC, Manassas, VA, USA; CRL-4011) were cultured in AEBM (ATCC; PCS-300-030) supplemented with the Bronchial Epithelial Cell Growth Kit (ATCC; PCS-300-040), 1% at 100 U/µg/mL Penicillin-Streptomycin (Gibco Life-Technologies, Waltham, MA, USA; 15140122), and 0.22% at 250 µg/mL Amphotericin-B (Gibco Life-Technologies; 15-290-018). VERO-E6 cells were cultured in High-Glucose DMEM (Gibco-Life Technologies; 12800-082). DMEM media was supplemented with Sodium Bicarbonate (Sigma-Aldrich; S5761-500G), 10% FBS (Gemini Bio-Products, West Sacramento, CA, USA; 100-106), 1% L-Glutamine (Gibco-Life Technologies; 25030-081), 1% HEPES (GE Healthcare Life Sciences, Chicago, IL, USA; SH30237.01), 1% at 100 U/µg/mL Penicillin–Streptomycin, and 0.22% at 250 µg/mL Amphotericin B. MRC-5 cells (ATCC; CCL-171) were cultured in EMEM (ATCC; 30-2003) supplemented with 10% FBS, 1% at 100 U/µg/mL Penicillin–Streptomycin, and 0.22% at 250 µg/mL Amphotericin B. HCT-116 cells were kindly provided by Dr. Narry Kim, Ph.D. (Seoul National University; Seoul, Republic of Korea), and were cultured in High-Glucose DMEM (Gibco-Life Technologies; 12800-082). DMEM media was supplemented with 1× MEM NEAA (Gibco-Life Technologies; 11140076) (Sodium Bicarbonate (Sigma-Aldrich; S5761-500G), 10% FBS (Gemini Bio-Products; 100–106), 1% L-Glutamine (Gibco-Life Technologies; 25030-081), 1% HEPES (GE Healthcare Life Sciences; SH30237.01), 1% at 100 U/µg/mL Penicillin–Streptomycin, and 0.22% at 250 µg/mL Amphotericin B.

All cell lines were cultured at 37 °C and 5% CO_2_ in humidified incubators, dissociated from plasticware with 0.25% Trypsin-EDTA (Gibco-Life Technologies; 25200114) with Trypsin-inactivation by 1% FBS in 1× DPBS (Corning, Corning, NY, USA; 20-031-CV-CL), and passed at ratios appropriate to the growth rate of each cell line. Experiments conducted in these cell lines were performed when cells were in the logarithmic phase of growth, at 60–70% confluency, and within 2 weeks after thawing from liquid nitrogen (2–3 passages). All cell lines were tested for Mycoplasma contamination and authenticated by ATCC using the Mycoplasma testing service and the Short Tandem Repeat profiling service.

### 2.3. SDS-PAGE and Western Blotting

Protein lysates were generated from cell pellets with the addition of 30–100 uL NP-40 Lysis Buffer [(50 mM Tris-HCl, pH 8.0, 150 mM NaCl, 1 mM EDTA, 1% NP-40, 0.1% SDS, 0.5 mM sodium metavanadate, Protease Inhibitor Cocktail (Sigma-Aldrich)]. Lysates were incubated on ice for 15 min and clarified by centrifugation for 5 min at 14,000× *g* and 4 °C. The concentrations of the protein extracts were determined using the Pierce BCA Protein Assay (ThermoFisher Scientific, Waltham, MA, USA; 23227), following the manufacturer’s protocol, with absorbance values plotted against an absorbance standard curve generated by a dilution series of BSA at 2 mg/mL (ThermoFisher Scientific; 23209). BCA Assay results were read using a BioTek Cytation 5 imager at 542 nm.

30–50 µg of total cell lysate protein was separated by SDS-PAGE on 10% Mini-PROTEAN TGX precast gels (Bio-Rad, Hercules, CA, USA; 4561033EDU) at 130 V for 1.5 h at 4 °C, followed by transfer to Immobilon-FL 0.45 μm PVDF membranes (Sigma-Aldrich, St. Louis, MO, USA; 05317) at 100 V for 2 h at 4 °C. The membranes were then blocked with 5% milk/PBS-T buffer (5% nonfat dried milk, 1× Dulbecco’s Phosphate-Buffered Saline, and 0.5% Tween-20) for 1 h at room temperature. Membranes were incubated overnight at 4 °C with primary antibodies in 5% milk/PBS-T buffer (1:1000) and then washed three times at room temperature for 10 min with PBS-T. The membranes were then incubated with the appropriate horseradish peroxidase (HRP)-conjugated secondary antibodies in 5% milk/PBS-T buffer (1:10,000) for 45 min at room temperature. The membranes were then washed three times for 10 min at room temperature in PBS-T. Proteins were detected using the Pierce SuperSignal West Pico Chemiluminescent Substrate (ThermoFisher Scientific; 34577) or Pierce SuperSignal West Femto Maximum Sensitivity Substrate (ThermoFisher Scientific; 34095) and imaged using an iBright image system/analysis software. Primary and Secondary antibodies used in this study can be found in Appendix A.

### 2.4. Viral Propagation, Quantification, and Infection

The following ancestral and variant SARS-CoV-2 stocks were deposited by the Centers for Disease Control and Prevention and obtained from BEI Resources, NIAID, NIH: SARS-Related Coronavirus 2, Isolate hCoV-19/USA-WA1/2020, NR-52281 (GISAID: EPI_ISL_404895.2); SARS-Related Coronavirus 2, Isolate hCoV-19/England/204820464/2020, NR-54000, contributed by Bassam Hallis (GISAID: EPI_ISL_683466); SARS-Related Coronavirus 2, Isolate hCoV-19/USA/MD-HP01542/2021 (Lineage B.1.351), in Homo sapiens Lung Adenocarcinoma (Calu-3) Cells, NR-55282, contributed by Andrew S. Pekosz (GISAID: EPI_ISL_890360). The following coronaviruses were obtained through BEI Resources, NIAID, NIH: Human Coronavirus, 229E, NR-52726; Human Coronavirus, OC43 in HRT-18G Cells, NR-56241. The following virus was obtained through BEI Resources, NIAID, NIH: Sindbis Virus, 80-2449, NR-51641. All virus stocks were propagated in the modified VERO E6-hACE-2/hTMPRSS-2 cell line, kindly gifted to us by Dr. Luis Martinez-Sorbido, University of Texas-Health San Antonio, as previously described, with the exception of HCoV-229E, which was propagated in the MRC-5 cell line [29]. VERO E6-hACE-2/hTMPRSS-2 or MRC-5 cells were plated in T-150 flasks (Corning; CLS431465), grown to 90% confluency, and infected with 500 uL stock virus. The virus was allowed to propagate for 72–96 h and observed daily for cytopathic effects (syncytia formation). Virus in the infection medium was collected and clarified to remove cell debris by centrifugation at 500× *g*, 4 °C, for 5 min. Stocks were aliquoted into gasketed vials (ThermoFisher-Scientific; 5000-1020) and stored at −80 °C until infection. Viral stocks were sequenced prior to use to ensure SARS-CoV-2 subvariant type and to ensure no additional mutations.

Viral stock titers were determined by viral plaque assay, and the generation of SARS-CoV-2 transcript copy number equivalent standard curves via quantitative RT-PCR, as previously described in [29]. Synthetic SARS-CoV-2 nucleocapsid gene RNA was synthesized for use in the standard curve and kindly gifted to us by Dr. Aaron Robart, West Virginia University. Viral plaque assays were performed in VERO E6-hACE-2/hTMPRSS-2 cells or MRC-5 cells, in 6-well plates, through the preparation of a stock dilution series, and the addition of methylcellulose overlays for plaque formation. Plaques were visualized via crystal violet staining and quantified by the following formula: Titer (PFU/mL) = (Number of Plaques Counted/0.2 mL) × 10Dilution Counted. qRT-PCR standard curve generation for SARS-CoV-2 stocks was performed by preparation of a dilution series of the SARS-CoV-2 synthetic N Gene Standard, cDNA generation using the iScript cDNA Synthesis Kit, and qPCR performed using the Sso-Advanced Universal SYBR Green Supermix. The sequence of the forward/reverse primer set specific to the SARS-CoV-2 nucleocapsid gene can be found in Appendix A. RNA transcript copies within each µL of the RNA standard were calculated as [(RNA concentration in ng/µL) × (Avogadro’s number)]/[(Molecular weight of transcript in g/mol) × 1,000,000,000] = RNA Copies/µL. Upon completion, each standard curve dilution was expected to fall within 3.3cT values, corresponding to one log10 copies.

Prior to infection, cells were seeded into 10 cm dishes and allowed to reach 70% confluency. After 24 h, cell lines were infected with SARS-CoV-2 subvariants, HCoV-229E, or HCoV-OC43 at a multiplicity of infection (MOI) of 1.0 (one viral particle per plated cell) or infected with Sindbis virus at an MOI of 5.0. 1 mL virus stocks were combined in a 15 mL conical tube, homogenized, and the appropriate amount of virus was added dropwise to each 10 cm, for the appropriate MOI. For mock-infected control samples, VERO E6-hACE-2/hTMPRSS-2 infection media–generated using the same viral propagation protocol without virus–was added to cells at a volume equal to the highest volume of virus stock used in the infected sample groups.

Two days (48 h) post-infection, cells were washed with ice-cold 1× DPBS, collected via scraping in 5 mL ice-cold 1× DPBS, and pelleted at 200× *g*/4 °C for 7 min. Cell pellets were washed once with ice-cold 1× DPBS, re-pelleted under the same conditions, and stored at −80 °C for RNA extraction or generation of protein lysates. All SARS-CoV-2 viral propagations and infections were performed under biosafety level 3 laboratory conditions.

### 2.5. Poly (I:C) Treatment

In total, 50 µg of synthetic polyinosinic–polycytidylic acid (Poly (I:C)) (Sigma Aldrich; 42424-50-0) in 500 µL 1× DPBS, or 500 µL 1× DPBS (mock-treated control), was transfected into cells seeded on 10 cm dishes. 24 h post-treatment, cells were collected and total RNA/protein lysates extracted.

### 2.6. Immunofluorescence and Signal Intensity Quantification

NuLi-1 cells were plated on poly-L-lysine-coated glass coverslips (Neuvitro Corporation, Camas, WA, USA; GG-25-1.5-PLL) at 5 × 10^4^ cells/coverslip in appropriate culture media infected with either mock infection media or an mCherry-tagged SARS-CoV-2 virus (kindly gifted to us by Dr. Luis Martinez-Sorbido, University of Texas-Health San Antonio) at an MOI of 1.0, and incubated at 37 °C and 5% CO_2_ in a humidified incubator for 48 h. Coverslips were then washed for 5 min, 3 times with ice-cold 1× DPBS and fixed with 4% paraformaldehyde (ThermoFisher Scientific; 28906) in 1× DPBS at a pH of 7.4, for 30 min at 4 °C. Samples were permeabilized by incubation with 0.1% Triton X-100 (Sigma Aldrich; X100-500ML) in 1× DPBS for 10 min at room temperature. Coverslips were washed three times for 5 min each with 1× DPBS and blocked using 1% BSA (Sigma Aldrich; 9048-46-8), 22.52 mg/mL glycine (Sigma Aldrich; G7126-500G), in 1× PBS-T (1× PBS + 0.1% Tween-20 (Sigma Aldrich; 655204-100ML)) for 30 min at room temperature. Samples were incubated with the appropriate diluted primary antibodies Appendix A in 1× PBS-T at room temperature for 1 h in the dark. Coverslips were washed for 5 min, 3 times with 1× DPBS, and incubated with the appropriate diluted secondary antibody Appendix A in 1% BSA in 1× PBS-T for 1 h in the dark. Samples were again washed 3 times with 1× DPBS. Cells were counterstained with NucBlue (Invitrogen; R37606), 2 drops in 1 mL 1× DPBS for 5 min at room temperature. Coverslips were briefly rinsed with 1× DPBS and mounted/sealed onto glass slides with VECTASHIELD PLUS Antifade Mounting Medium (Vector Laboratories, Newark, CA, USA; H-1900) mounting media. Slides were imaged using a Nikon A1R confocal microscope. Analysis and statistical testing were performed using Cell Profiler V4.2.8, Microsoft Excel 2024, and GraphPad Prism V10.

### 2.7. Transient Transfection of siRNAs

NuLi-1 cells were transiently transfected using Lipofectamine RNA-iMAX (Invitrogen; 13778150) according to the manufacturer’s instructions. In 6-well plates, per well, 9 µL Lipofectamine RNAi-MAX and 200 pmol appropriate Dharmacon custom siRNA or scramble control Appendix A were diluted in 250 μL Opti-MEM medium (Gibco Life-Technologies; 31985-070). The RNA-iMAX and siRNA were combined and allowed to incubate together for 1 h at room temperature. The mixture was added, dropwise, to the appropriate well. siRNAs were allowed to incubate with cells for 24 h prior to infection experiments, and 72 h total prior to cell pelleting and RNA/protein lysate extractions. Lipofectamine RNA-iMAX and siRNA concentrations were optimized, and equal volumes/amounts were used across both cell lines.

### 2.8. Tiling RT-PCR and Exon Amplification RT-PCR

Primer sets overlapping the entire Drosha open reading frame, and Drosha exon-group-specific primer sets were designed Appendix A. cDNA synthesis was performed using the iScript Select cDNA Synthesis Kit (BioRad, Hercules, CA, USA; 170-8897). RT-PCR was performed using the Platinum SuperFi II PCR Master Mix (Invitrogen; 12368010). PCR products were visualized via gel electrophoresis, using 0.5–2.0% agarose gels containing 0.5 µg/mL Ethidium Bromide, at 10 V/cm^2^ for 2 h. Gels were imaged using an iBright image system/analysis software (Version 1.8.1).

### 2.9. Statistical Analysis

All statistical analysis is presented as mean ± SD (ns, no significance; *, *p* < 0.05; **, *p* < 0.01; ***, *p* < 0.001; ****, *p* < 0.0001). The sample data Gaussian distribution was confirmed using the Shapiro–Wilk normality test. The statistical significance between groups was determined using Student’s *t*-Test, unless comparing three or more groups, in which case samples were analyzed using a two-way ANOVA, followed by Tukey’s multiple comparisons test. Analyses were performed using GraphPad Prism V10 and Microsoft Excel 2024.

## 3. Results

### 3.1. SARS-CoV-2 Infection Induces Drosha Cleavage into Distinct Isoforms

Previous work in our laboratory demonstrated that a smaller isoform of Drosha, expressed during cellular quiescence, localizes to the cytoplasmic fraction [28]. Additional studies have linked this translocation to cellular stress and viral infection [25,30,31]. Several Drosha isoforms exist under homeostatic conditions, arising from both alternative splicing and proteolytic cleavage [25,26,27]. Motivated by these findings, we characterized the expression of three distinct Drosha isoforms during SARS-CoV-2 infection (Figure 1A).

Notably, no clear change in the protein expression of full-length or cleaved P140 Drosha expression was observed following infection with SINV or treatment with the viral mimetic poly(I:C) (Figure 1B). Similarly, infection with two non-severe human coronaviruses, HCoV-OC43 and HCoV-229E, did not alter Drosha isoform expression. This observation is particularly significant for HCoV-OC43, a beta-coronavirus like SARS-CoV-2 [32]. This was seen despite strong infection/treatment efficacy, as seen by the expression of several virus-specific and interferon-specific gene targets (Figure 1C). However, SARS-CoV-2 infection uniquely induced a clear reduction in full-length Drosha and increased expression of P140 and P25 isoforms (Figure 1D).

This isoform shift was consistent across several SARS-CoV-2 variants, with infection validated via qPCR of three viral gene targets (Figure 1D,E). The identity of all three protein bands (full-length, P140, P25) was confirmed through siRNA knockdown Appendix A. While Dicer expression increased significantly following SARS-CoV-2 infection, no other miRNA biogenesis proteins exhibited altered expression at either the protein or transcript level Appendix A. Collectively, these findings suggest that SARS-CoV-2 specifically alters Drosha’s isoform expression via proteolytic processing, distinct from the effects observed with other RNA viruses or non-severe coronaviruses.

### 3.2. Alternative Splicing Variation Remains Unchanged Following SARS-CoV-2 Infection

Drosha is regulated both transcriptionally and post-translationally, with known sensitivity to cellular stress [33,34]. Splice variants affecting Drosha’s subcellular localization have been described, particularly involving the nuclear localization signal (NLS) and intron retention [26,27]. Given that SARS-CoV-2 manipulates host RNA-binding proteins and splicing mechanisms [12], we sought to distinguish between Drosha cleavage and alternative splicing as the cause of isoform shifts.

Using multiple primer sets spanning the Drosha transcript Appendix A, we found no overall change in Drosha mRNA expression following infection (Figure 2A). Amplicons covering ~1500 bp regions revealed no evidence of altered transcript structure (Figure 2B). Targeted PCR of exons 1–3 showed decreased expression of the Drosha-1 variant, but this did not reflect a broader shift in cytoplasmic versus nuclear isoform expression (Figure 2C,D).

We further analyzed splicing events in regions encoding Drosha’s NLS (exons 5–8), the proline-rich domain (exons 1–3), and the double-stranded RNA-binding domain (exons 31–35) and observed no infection-associated differences (Figure 2D). Intron retention events between exons 27–28 and 32–33—previously shown to produce altered Drosha protein isoforms—were also unaffected by infection [27] (Figure 2E). Together, these results indicate that the observed changes in Drosha isoforms following SARS-CoV-2 infection are not due to alternative splicing or intron retention.

### 3.3. Drosha Cleavage Isoforms Show Altered Subcellular Localization Following Infection

Under normal conditions, Drosha is predominantly nuclear, though it may translocate to the cytoplasm during cellular stress or viral infection [25,30,31]. Phosphorylation of serine/threonine residues—particularly those preceding proline—regulates protein localization [35], and Drosha’s nuclear import is controlled by N-terminal serine phosphorylation via GSK3B [36,37]. As SARS-CoV-2 disrupts host phosphorylation and RNA transport, we hypothesized altered Drosha localization following infection [12,38,39,40].

Using an mCherry-tagged SARS-CoV-2 strain (WA1–USA/2020) and an antibody recognizing full-length and P140 Drosha (Figure 3A), we observed an overall reduction in Drosha protein and a corresponding decrease in both cytoplasmic and nuclear mean fluorescence intensity (MFI) (Figure 3A,B). However, Drosha distribution shifted notably toward the cytoplasm, with decreased nuclear localization following infection (Figure 3C). These results suggest that while overall Drosha expression declines, its relative enrichment in the cytoplasm is enhanced in response to SARS-CoV-2.

### 3.4. Drosha Knockout Leads to a Decrease in the Expression of SARS-CoV-2 Genomic and Sub-Genomic RNA

Drosha translocation to the cytoplasm has been reported for multiple viral infections, including those involving diverse genome structures [31,41]. In addition to its role in microRNA processing, Drosha has been implicated in regulating viral gene expression in KSHV [42,43]. SARS-CoV-2 interacts with host RNA-regulatory proteins, and Drosha localization to replication-associated compartments may influence viral RNA processing [44].

To examine Drosha’s role in SARS-CoV-2 replication, we compared wild-type and Drosha-knockout HCT-116 cells following infection (Figure 4A). We used qPCR primer sets targeting key regions across the SARS-CoV-2 genome (Figure 4B and Appendix A). Coronaviruses generate sub-genomic RNAs via discontinuous transcription, and expression varies by gene location along the genome [45]. For example, the nucleocapsid (N) gene, located near the 3′ end, exhibits higher expression than upstream genomic regions [29,46].

In Drosha-knockout cells, we observed a significant decrease in N gene expression, consistent with reduced viral replication (Figure 4C). Expression of ORF1a and 5′UTR regions also declined, indicating a broader reduction in genomic RNA abundance. These findings demonstrate that Drosha is required for efficient expression of both SARS-CoV-2 genomic and sub-genomic RNAs.

## 4. Discussion

Through the data presented here, we have shown that the Drosha P140 isoform is enriched in cells following SARS-CoV-2 infection, accompanied by a decrease in the canonical full-length isoform—a shift that appears unique to SARS-CoV-2. This isoform alteration occurs in tandem with Drosha’s translocation to the cytoplasm. Both phenomena impact SARS-CoV-2 replication, as Drosha ablation via a CRISPR-based knockout model led to decreased expression of viral genomic and sub-genomic RNA.

Drosha isoform variation has previously been attributed to both alternative splicing and proteolytic cleavage. However, our findings indicate that the isoform shift observed during SARS-CoV-2 infection results from proteolytic cleavage, independent of splicing mechanisms [25,26,27]. This is supported by Drosha-specific siRNA knockdown and RT-PCR experiments targeting exon groups associated with the nuclear localization signal. Prior studies have shown Drosha cleavage in response to cellular stressors such as heat, mediated by caspases or calpains activated via phosphorylation pathways like p38 MAPK [33,34]. SARS-CoV-2 has been reported to alter the expression and activity of these proteases and associated cofactors [47,48,49].

SARS-CoV-2 may also facilitate Drosha cleavage through its viral proteases, 3CLpro or Mpro, which are serine proteases known to target host substrates. These enzymes could potentially cleave residues within Drosha’s N-terminus. In parallel, the observed cytoplasmic redistribution of Drosha may result from disruptions in phosphorylation pathways, particularly those involving GSK3β. SARS-CoV-2 is known to recruit GSK3β for phosphorylation of its nucleocapsid protein [39,50], and GSK3β also phosphorylates Drosha’s N-terminal serine residues to regulate its nuclear localization [36,37].

This phenomenon of nuclear export is likely SARS-CoV-2-specific, as comparable shifts in Drosha isoform expression were not induced by HCoV-OC43 or HCoV-229E. These findings support the hypothesis that SARS-CoV-2 uniquely manipulates Drosha localization and processing to facilitate replication.

The stress-induced nuclear export of Drosha observed here is consistent with previous studies documenting similar localization changes under stress conditions [30,31]. This cytoplasmic Drosha may influence viral replication through several mechanisms. In Drosha knockout cells, we observed a significantly reduced expression of SARS-CoV-2 ORF1a, N, and 5′UTR sequences, suggesting a supportive role for Drosha in viral replication. While the precise mechanism remains unresolved, Drosha may interact with viral RNA directly—through cleavage or structural modulation—or indirectly by altering host transcript or microRNA profiles [42,43,51,52].

Although our findings demonstrate a strong association between Drosha isoform expression and SARS-CoV-2 replication, we did not distinguish between the functional contributions of full-length, P140, and P25 isoforms. Future studies should aim to characterize isoform-specific roles and identify the host or viral proteases responsible for cleavage. Further mechanistic investigation will clarify whether Drosha acts through direct interaction with viral RNA or indirectly via host regulatory pathways.

Overall, our results suggest a non-canonical, cytoplasmic function for Drosha during SARS-CoV-2 infection—distinct from its established nuclear role in microRNA biogenesis. This highlights a novel mechanism by which SARS-CoV-2 may co-opt host RNA-processing enzymes to support its lifecycle. A deeper understanding of how RNA viruses manipulate host RNA regulatory networks will be essential in the development of novel antiviral strategies. Furthermore, the development of RNA vaccines against SARS-CoV-2 demonstrates not only the importance of RNA as a therapeutic tool but also the importance of understanding RNA processing pathways in disease, including COVID-19.

## Figures and Tables

**Figure 1 genes-16-01239-f001:**
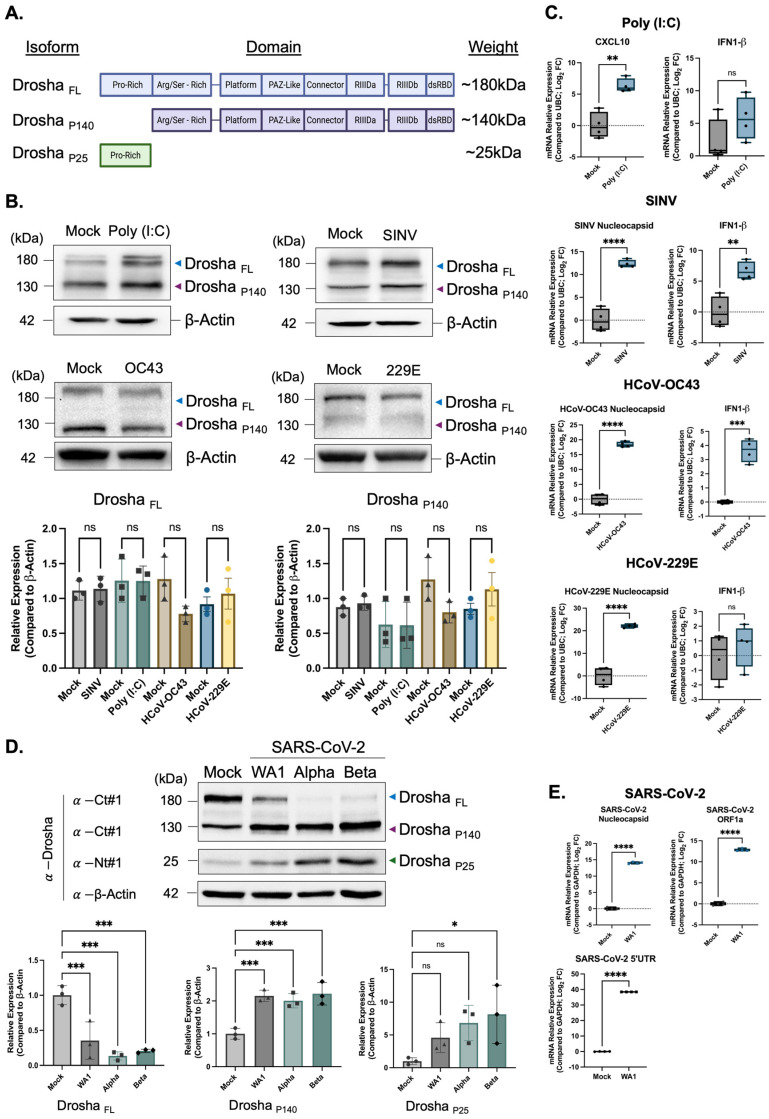
Drosha undergoes cleavage into smaller isoforms following infection by SARS-CoV-2 but not following treatment with poly (I:C), or infection by another positive-sense RNA virus/other non-severe human coronaviruses. (**A**) Schematic representation of Drosha full-length and proteolytically cleaved isoforms (P140, P25), with included domains and respective molecular weights. (**B**) Western blot analysis of Drosha protein isoforms in NuLi-1 cells 48 h post-treatment/infection with poly (I:C), Sindbis virus (SINV) (MOI 5.0), HCoV-229E (MOI 1.0), HCoV-OC43 (MOI 1.0). (**C**) Quantitative RT-PCR analysis of infected NuLi-1 cells, measuring expression of viral targets or antiviral response genes. Poly(I:C)-treated cells show elevated CXCL10 and IFN-Beta expression compared to the mock control group. Cells infected with SINV, HCoV-OC43, or HCoV-229E exhibit robust expression of viral nucleocapsid genes and IFN-β. (**D**) SARS-CoV-2 variants (WA1, Alpha, Beta). Representative immunoblots show full-length Drosha (FL), the P140 cleavage product, and a lower molecular weight isoform (P25), with β-actin used as a loading control. Bands corresponding to Drosha cleavage products are present following infection with each coronavirus but absent or faint in poly(I:C), SINV, HCoV-OC43, and HCoV-229E-treated/infected samples. (**E**) SARS-CoV-2-infected cells (WA1, Alpha, Beta) (MOI 1.0) display SARS-CoV-2 ORF1a, N gene, and 5′UTR expression. Data are normalized to Beta-Actin or GAPDH and are presented as mean fold change ± SD from at least three independent experiments. The statistical significance between groups was determined using a two-tailed Student’s *t* test, or two-way ANOVA followed by Tukey’s post hoc test (ns, not significant; * *p* < 0.05; ** *p* < 0.01; *** *p* < 0.001; **** *p* < 0.0001).

**Figure 2 genes-16-01239-f002:**
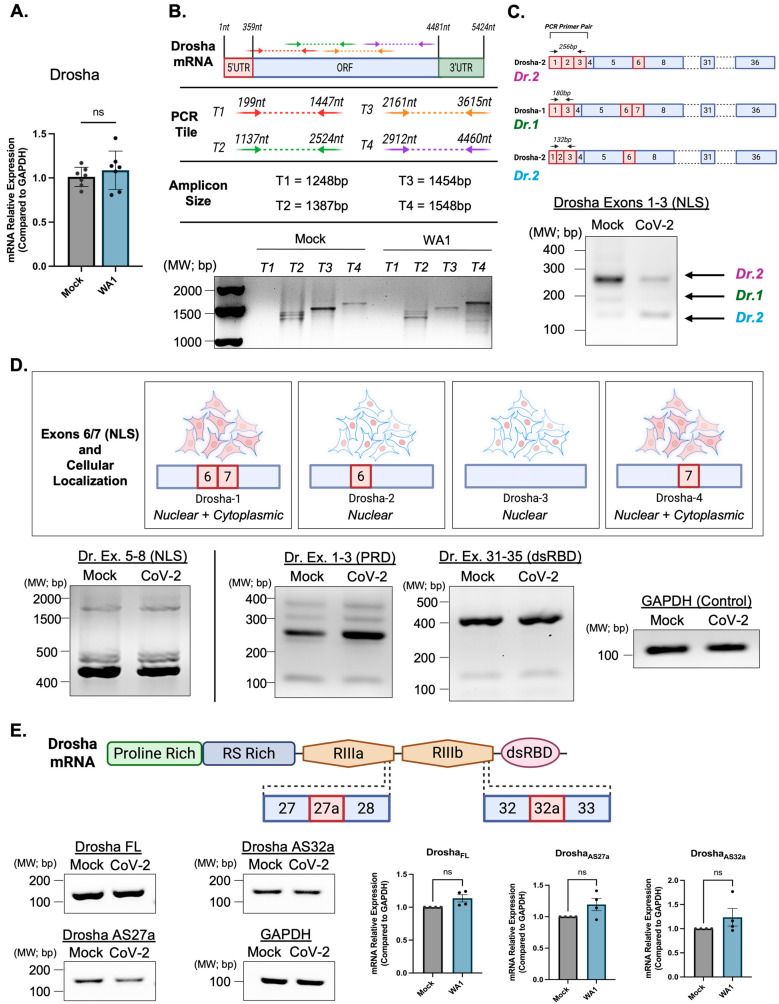
Drosha splice variant expression is largely unaltered at the transcript level following SARS-CoV-2 infection. (**A**) qRT-PCR shows no significant change in the expression of Drosha at the mRNA transcript level. (**B**) Agarose gel electrophoresis showing amplification of overlapping tiled regions spanning the full-length Drosha transcript (Tiles 2–4) in mock and SARS-CoV-2-infected NuLi-1 cells. No major differences in band intensity or size were observed, suggesting stable global Drosha transcript expression following infection. (**C**) Exon 1-3 amplification relating to exon 6/7 retention. Following SARS-CoV-2 infection, there is no clear trend in the loss or retention of exon 6, relating to the nuclear localization of Drosha. (**D**) RT-PCR analysis of Drosha exon groups in mock- and SARS-CoV-2 WA1-infected NuLi-1 cells. Primer sets targeted exon regions 1–3, 5–8, and 31–35. PCR products show comparable size and intensity across conditions, indicating no large-scale exon exclusion or inclusion. GAPDH amplification was included as a loading and cDNA quality control. (**E**) qRT-PCR analysis of potential intron retention events between exons 27/28 and 32/33 of the Drosha transcript. Minor band shifts were observed but did not indicate strong intron retention following infection. GAPDH amplification was included as a loading and cDNA quality control. Data are shown as mean ± SD of at least three biological replicates. The statistical significance between groups was determined using a two-tailed Student’s *t*-test (ns, not significant). Molecular weight markers are shown in base pairs (bp). Graphical figures were created using BioRender.com.

**Figure 3 genes-16-01239-f003:**
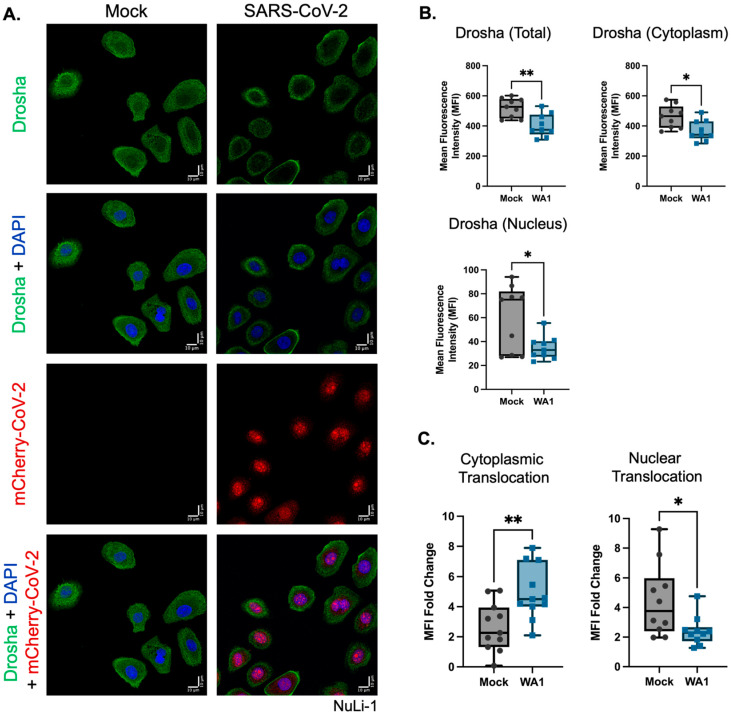
Following SARS-CoV-2 infection, there is a decrease in total Drosha expression and an increase in the cytoplasmic localization of Drosha in NuLi-1 cells. (**A**) Representative immunocytochemistry (ICC) images of mock and SARS-CoV-2-infected NuLi-1 cells stained for Drosha (green), DAPI (blue), and mCherry-tagged SARS-CoV-2 (red). (**B**) Composite ICC image analysis highlighting an overall decrease in Drosha expression following SARS-CoV-2 infection (MOI 1.0) in whole cells, as well as a decrease in both the nucleus and cytoplasm. (**C**) Quantification of the translocation of Drosha from both the cytoplasm and nucleus. Following infection by SARS-CoV-2 (MOI 1.0), there was an increase in the translocation of Drosha to the cytoplasm of the cell, accompanied by a decrease in the translocation of Drosha to the nucleus of the cell. Total, nuclear, and cytoplasmic Data shown are mean Drosha fluorescence intensities ± SD across at least 8 biological replicates and were quantified using CellProfiler. The statistical significance between groups was determined by using a two-tailed Student’s *t*-test (* *p* < 0.05; ** *p* < 0.01).

**Figure 4 genes-16-01239-f004:**
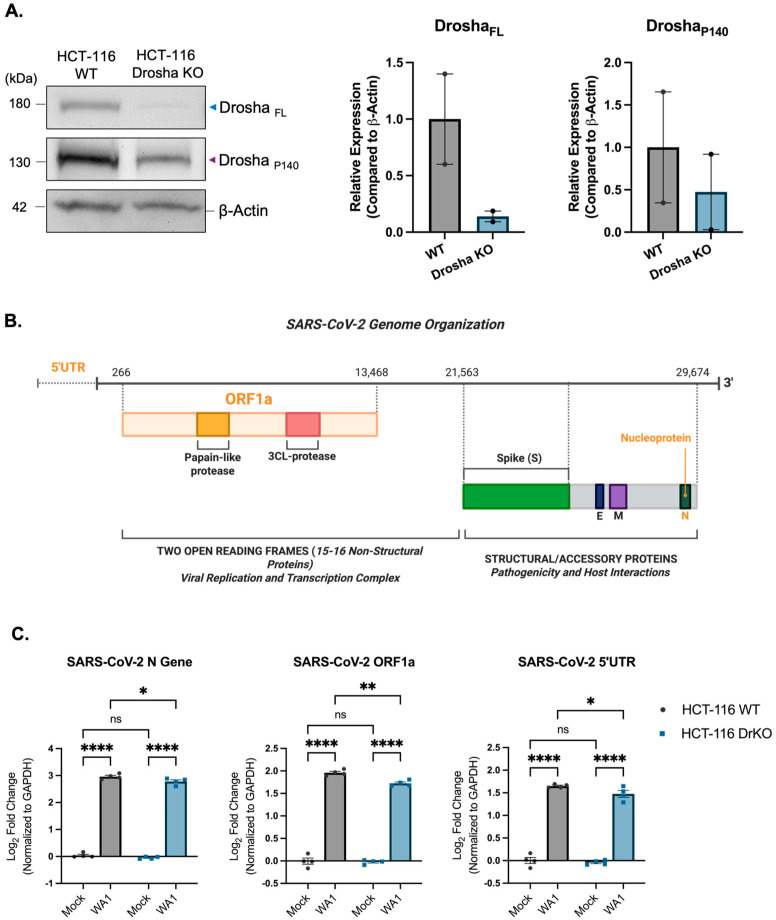
Drosha knockout leads to a decrease in the expression of SARS-CoV-2 genomic and sub-genomic RNA. (**A**) Representative Western blot image and analysis of full-length and P140 Drosha expression in HCT-116 wild-type (WT) and Drosha knockout (DrKO) cells. Data are normalized to Beta-Actin and are presented as mean fold change ± SD of two biological replicates. (**B**) Diagram of SARS-CoV-2 genome organization, with qPCR gene targets highlighted in orange. (**C**) Quantitative RT-PCR analysis of SARS-CoV-2 replication markers in HCT-116 WT and HCT-116 DrKO cells 48 h post-infection with SARS-CoV-2 WA1 (MOI 1.0). Transcript levels for Nucleocapsid (N), ORF1a, and 5′UTR regions show a significant decrease in expression, following SARS-CoV-2 infection, in Drosha knockout cells compared to WT. Data shown are normalized to GAPDH and are presented as mean fold change ± SD from three biological replicates. The statistical significance between groups was determined by using a two-way ANOVA followed by Tukey’s post hoc test (ns, not significant; * *p* < 0.05; ** *p* < 0.01; **** *p* < 0.0001). Graphical figures were created using BioRender.com.

## Data Availability

The original contributions presented in this study are included in the article/Appendix A. Further inquiries can be directed to the corresponding author.

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
