# Peer review of "SARS-CoV-2 Infection of Lung Epithelia Leads to an Increase in the Cleavage and Translocation of RNase-III Drosha; Loss of Drosha Is Associated with a Decrease in Viral Replication"

_genes, 2025, doi:10.3390/genes16101239_

Round 1

Reviewer 1 Report

Comments and Suggestions for Authors

In this study, Winters et al evaluated the effects of SARS-CoV-2 infection on the expression and localization of Drosha isoforms in immortalized human lung cells. These findings have implications for how the virus may manipulate the production of RNA processing enzymes to enhance viral load. According to the authors, several of the findings in this manuscript are distinct to SARS-CoV-2 and are not observed in other related viruses, which could have implications for treating severe coronaviruses. The science and reporting appear robust and the study represent a useful step forward for the field.

Abstract

  • The Methods section of the abstract does not describe the experimental design, only analysis methods. What was the experimental model?

Introduction

  • “SARS-CoV-2 has caused nearly more than 1.23 million deaths” (lines 36-37) – please clarify language
  • Great introduction

Methods

  • Thorough methods description and excellent reagent validation

Results

  • “Notably, clear no changes in full-length or cleaved P140 Drosha expression” (line 270) – please clarify language
  • Fig 1A is not referenced in the text
  • Fig 2 appears to be low-resolution and some of the diagrams are hard to read (panels B, C, D)

Discussion

  • Great discussion

Reviewer 2 Report

Comments and Suggestions for Authors

The pandemic of COVID induced by SARS-CoV-2 virus has shown us our lack of molecular biology knowledge and, moreover, the defencelessness of the medical protection system. As the authors correctly underlined during this sorrowful period, more than 1.2 million human beings died. Therefore, the new points of therapy, including identification/diagnosis, as well as protection, are highly demanded. The development of this area will bring us closer to a safer world in the event of a new pandemic. In the reviewed article entitled SARS-CoV-2 Infection of Lung Epithelia Leads to an Increase in the Cleavage and Translocation of RNase-III Drosha; Loss of Drosha is Associated with a Decrease in Viral Replication, the authors discussed the new finding, i.e., the microRNA-processing enzyme Drosha. The identification of the new role of this enzyme can be helpful in virus infection therapy and prognosis. The experimental techniques have been correctly used and described; moreover, the negative control was done, „Drosha knockout leads to a decrease in the expression of SARS-CoV-2 genomic and sub-genomic RNA” which is really important for such studies. The manuscript, as was expected, is well written and readable. The references were correctly selected. It would be wonderful if authors included some information about the vaccines developed and used during the pandemic.

The graphical data presentation is correct.

Reviewer 3 Report

Comments and Suggestions for Authors

The manuscript of Michael T Winters et al. reports the changes in Drosha isoform expression following infection with multiple SARS-CoV-2 variants. In addition, a distinct alteration in Drosha’s cellular localization post SARS-CoV-2 infection was observed.

The introduction provides important and detailed information on experimental system. The methods are clearly described, specifying details of the conditions under which the experiments were performed. The hypothesis of the work was corroborated.

            In my opinion the manuscript is of interest but minor questions, reported below, need to be clarified.

1- Figure 2, up of panel B: The authors need to increase image definition. The same thing must be done for the panel D.

2- Please check the citation at line 304 and if correct, align the citation according to the journal's instructions.

3- I wonder if it would be possible to prove the different cellular localization of Drosha by immunoprecipitation and mass spectrometry (in order to also verify the type of isoform (paragraph 3.3).
